# Evaluating the clinical effectiveness of the NHS Health Check programme: a prospective analysis in the Genetics and Vascular Health Check (GENVASC) study

Radoslaw Debiec ,[1] Daniel Lawday,[1] Vasiliki Bountziouka,[1,2] Emma Beeston,[1] Chris Greengrass,[1] Richard Bramley,[1] Sue Sehmi,[1] Shireen Kharodia,[1] Michelle Newton,[1] Andrea Marshall,[1] Andre Krzeminski,[3] Azhar Zafar,[4,5] Anuj Chahal ,[6] Amardeep Heer,[7] Kamlesh Khunti,[4] Nitin Joshi,[8] Mayur Lakhani,[9] Azhar Farooqi,[9] Riyaz Patel ,[10] Nilesh J Samani[1]

**Correspondence to**
Dr Radoslaw Debiec;
rmd24@leicester.ac.uk

## ABSTRACT

**Objective** The aim of the study was to assess the clinical effectiveness of the national cardiovascular disease (CVD) prevention programme—National Health Service Health Check (NHSHC) in reduction of CVD risk.

**Design** Prospective cohort study.

**Setting** 147 primary care practices in Leicestershire and Northamptonshire in England, UK.

**Participants** 27 888 individuals undergoing NHSHC with a minimum of 18 months of follow-up data.

**Outcome measures** The primary outcomes were NHSHC attributed detection of CVD risk factors, prescription of medications, changes in values of individual risk factors and frequency of follow-up.

**Results** At recruitment, 18% of participants had high CVD risk (10%–20% 10-year risk) and 4% very high CVD risk (>20% 10-year risk). New diagnoses or hypertension (HTN) was made in 2.3% participants, hypercholesterolaemia in 0.25% and diabetes mellitus in 0.9%. New prescription of stains and antihypertensive medications was observed in 5.4% and 5.4% of participants, respectively. Total cholesterol was decreased on average by 0.38 mmol/L (95% CI −0.34 to −0.41) and 1.71 mmol/L (−1.48 to −1.94) in patients with initial cholesterol >5 mmol/L and >7.5 mmol/L, respectively. Systolic blood pressure was decreased on average by 2.9 mm Hg (−2.3 to −3.7), 15.7 mm Hg (−14.1 to −17.5) and 33.4 mm Hg (−29.4 to −37.7), in patients with grade 1, 2 and 3 HTN, respectively. About one out of three patients with increased CVD risk had no record of follow-up or treatment.

**Conclusions** Majority of patients identified with increased CVD risk through the NHSHC were followed up and received effective clinical interventions. However, one-third of high CVD risk patients had no follow-up and therefore did not receive any treatment. Our study highlights areas of focus which could improve the effectiveness of the programme.

**Trial registration number** NCT04417387.

## STRENGTHS AND LIMITATIONS OF THIS STUDY

⇒ Prospective observation of large cohort of patients from 147 primary care practices undergoing National Health Service Health Check (NHSHC).

⇒ Purposefully built, comprehensive database.

⇒ Possible selection bias—individuals volunteering to take part in the study might represent more 'proactive' subgroup of NHSHC attendees.

⇒ Larger than national average proportion of non-white ethnicities representing population structure of the region.

⇒ The performed analysis focused on cardiovascular disease risk factors recorded in the primary care record but has not covered influence of NHSHC on lifestyle: physical exercise, diet or alcohol consumption.

## INTRODUCTION

Cardiovascular disease (CVD) is a leading cause of premature morbidity and mortality in England.[1 2] To address this, in 2009 the UK government introduced first in the world rolling national CVD prevention programme, delivered by the National Health Service (NHS). Now known as the NHS Health Check (NHSHC), the programme aims to identify and treat the main risk factors driving CVD, among people aged 40–74 years, who were otherwise well.[3] It was estimated through economic modelling that NHSHC could prevent over 1600 myocardial infarctions and strokes and 650 premature CVD deaths annually.[4]

The absence of randomised clinical trials demonstrating the effectiveness of such a programme, has continued to raise debate about its value.[5 6] Instead, several studies

sought to assess the effectiveness of the NHSHC using various observational datasets, but often with conflicting results, casting further doubt about the ability of the programme to achieve the stated goals.[7–11]

The Genetics and Vascular Health Check study (GENVASC) was instigated in 2012 to assess the value of adding a polygenic risk score (PRS) for CVD to the NHSHC for prediction of cardiovascular events. The study recruited patients attending for the NHSHC in 147 primary care practices in two counties in England (Leicestershire and Northamptonshire) and collected comprehensive clinical data comprising the NHSHC as well as subsequent follow-up visits. This dataset now permits evaluation of the impact of NHSHC in more detail than previously reported. In particular, the dataset allows for detailed analysis of the sequelae of detecting a risk factor, prescription of pharmacotherapy, clinical follow-up and their correlation with observed reduction in the risk factor. Specifically, we were able to fully characterise the subpopulation of individuals, who attended the initial NHSHC and were identified to have high and very high cardiovascular risk as well as individuals with gross elevations of individual risk factors. The main aim of our study was to assess effectiveness of clinical interventions and identify factors that may influence clinical and cost effectiveness of the NHSHC.

## METHODS
### National Health Service Health Check
The NHSHC is performed during a dedicated primary care visit and involves structured assessment of CVD risk.[12] Adults aged 40–74 years, without pre-existing CVD, diabetes mellitus, hypertension (HTN), hypercholesterolaemia and not taking statins are invited to attend the programme. The 10-year risk of developing CVD is calculated using QRISK2 score.[13] The attendees receive consultation regarding their CVD risk and individual risk factors: body mass index (BMI), tobacco smoking, blood pressure (BP), cholesterol level, alcohol intake, physical activity and risk of diabetes. Patients found to have increased overall CVD risk (QRISK2 score ≥10%) or elevated individual risk factor values are given lifestyle modification advice, pharmacological treatment[14] and/or are referred to specialist services. Subjects with QRISK2<10% are invited for re-assessment every 5 years.

### GENVASC study
GENVASC is a large observational study run in conjunction with Clinical Commissioning Groups and Primary Care practices in UK and coordinated by the NIHR Leicester Biomedical Research Centre. The GENVASC study recruited multiethnic individuals aged 40–74 years, attending the NHSHC at any of the 147 participating primary care practices in Leicestershire and Northamptonshire, UK (online supplemental figure 1). The main aim of the study is to evaluate the additional clinical value of a PRS on top of traditional risk scores in prediction of

subsequent CVD events.[15] Participants of the GENVASC are followed up using primary care databases. The collected information includes clinical diagnoses, laboratory and imaging tests, hospital admissions and referrals to external services as well as medicinal prescriptions.

### Study participants
For the purpose of current analysis we included 27 888 participants of the GENVASC study recruited between 17 June 2010 and 04 September 2018 who had 18 months of follow-up data.[15] Based on an estimate from Patel *et al*[16] about the annual attendance to NHSHC in Leicestershire and Northamptonshire, the population used for current analysis constituted around 10% of all NHSHC attendees in these two counties. All participants gave their consent to access their medical records during the duration of the study.

### Clinical variables and definitions
For details regarding methods of assessment and extraction of clinical and laboratory variables (see online supplemental material). Overall, 10-year CVD risk was defined from QRISK2 score and categorised as low (<10%), high (10–20%) and very high (>20%).[13] BP was defined as normal if systolic blood pressure (SBP) <140 mm Hg and/or diastolic blood pressure (DBP) <90 mm Hg; grade 1 HTN if SBP was 140–159 mm Hg and/or DBP 90–99 mm Hg, grade 2 HTN if SBP was 160–179 mm Hg and/or DBP 100–109 mm Hg, grade 3 HTN if SBP was ≥180 mm Hg and/or DBP≥110 mm Hg.[17] Total cholesterol (TCh) was defined as normal for values <5.0 mmol/L, high for TCh values 5.0–7.49 mmol/L and very high for TCh≥7.5 mmol/L.[18] Non-high density lipoprotein (non-HDL) was defined as normal if values were <3.8 mmol/L and high for values ≥3.8 mmol/L.[18] Kidney function was considered normal if estimated glomerular filtration rate (eGFR) ≥60 mL/min/1.73m²; chronic kidney disease (CKD) was diagnosed if eGFR<60 mL/min/1.73m².[19] Body weight was assessed using BMI and considered as healthy body weight if BMI<25 kg/m² (<23 kg/m² for South Asians), overweight for BMI from 25 to 29.9 kg/m² (23 to 27.4 kg/m² for South Asians) and obese if BMI≥30 kg/m² (>27.5 for South Asians).

Baseline clinical observations were defined as measurements taken during the NHSHC or the nearest measurement taken around the date of NHSHC but within 30 days from the date of the check. Any new diagnoses of CVD risk factors made from the date of NHSHC to 12 weeks after were attributed to the NHSHC visit. This 12-week period, although arbitrary, was considered appropriate for making clinical diagnosis and starting any treatment and supported by the bar plot of distribution of weekly diagnoses post-NHSHC (figure 1).

### Outcomes of interest
The assessment of the impact of the NHSHC was performed using the following measures:

# Weekly Incidence of CVD risk factors in relation to NHS Health Check

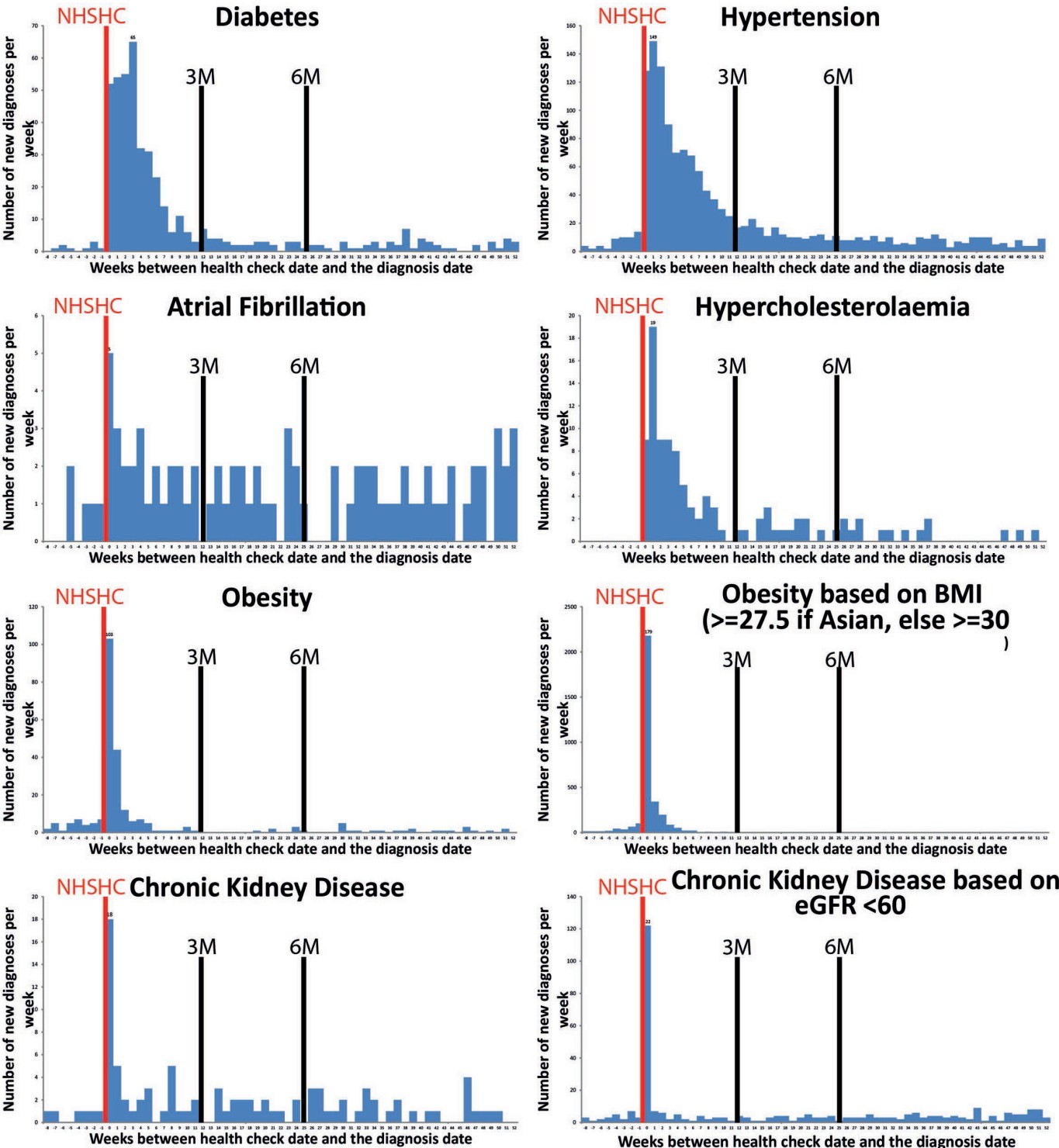

**Figure 1** Weekly incidence of cardiovascular disease risk factors in relation to National Health Service Health Check (NHSHC). BMI, body mass index; eGFR, estimated glomerular filtration rate.

► Risk factor detection—absolute number and proportion of participants with abnormal values of BP, TCh, eGFR, BMI, current tobacco smoking or having a clinical diagnosis of HTN, diabetes, hypercholesterolaemia, CKD or atrial fibrillation. Both the rates of clinically coded diagnoses and the rates of observed abnormal values were presented to accurately assess prevalence of risk factors.

- ► Medication prescription and non-medical interventions (eg, smoking cessation)—absolute number and proportion of subjects started on a given therapy.
- ► Frequency of follow-up—number of clinical visits/reviews within the specified periods.
- ► Change in values of individual risk factors—difference between the value of a given clinical characteristic recorded during the NHSHC and first value of this characteristic recorded after 12 months from the NHSHC visit.

## Data extraction

Data about participants were extracted from the primary care information technology systems. The primary care databases (EMIS and SystmOne) were used to obtain participants' sociodemographic information (age, sex and ethnicity), anthropometric measurements (height, weight), information about health-related behaviour (tobacco smoking), clinical diagnoses, blood results and prescribed medicines. Relevant data were extracted using NHS Numbers, SNOMED CT/DM&D and Clinical Terms Version 3 (formerly known as the Read codes).

## Statistical analysis

Categorical variables (sex, ethnicity, deprivation based on quintiles of Townsend 2011 index (for further information on Townsend index see Supplementary Data—Townsend Deprivation Score), smoking status, BP category, TCh category, renal function category, overweight/obesity, QRISK2 category) are shown as frequencies (relative frequency) while continuous data are shown as means and SD if normally distributed (SBP, DBP, TCh), or median (IQR) otherwise (age, eGFR). Normality was assessed using graphical methods (ie, histograms, P–P and Q–Q plots).

Group comparisons were performed using a $\chi^2$ test for categorical variables and the one-way analysis of variance for the normally distributed continuous variables or the Kruskall-Wallis for the non-normally distributed ones. The change in the levels of individual risk factors was calculated as the difference between the initial NHSHC measurement and the measurement during follow-up, after a period of 12 months. To assess the statistical significance of that comparison, the paired t-test for the continuous and the McNemar test for the nominal data were used.

Subjects with no repeated values recorded for the measured variables from after the initial NHSHC visit to the total time of follow-up observation of 18 months were considered as not followed up. The comparisons between individuals without follow-up and those that were followed up were performed with the independent t-test and Pearson's $\chi^2$ test for the continuous and nominal data, respectively. A logistic regression model was used to assess whether sociodemographic (age, sex, deprivation, ethnicity) and clinical factors (SBP, DBP, TCh, BMI, smoking) were associated with the absence of follow-up. Significance level was set to 0.05, all tests

were two-sided and Stata V.16.0 was used for all the calculations.[20]

## Patient and public involvement

GENVASC was developed in collaboration with the NIHR Leicester Biomedical Research Centre Cardiovascular Public Involvement Group. The research plan and all public facing materials were reviewed and approved prior to ethics submission. The group was updated biannually on the study. This was later extended to our tissue public involvement group (The Exceed Group).

## RESULTS

### Key characteristics of the study cohort

Overall 27 888 GENVASC participants were included in the current analysis. The median age of included participants was 51 years, 44% (n=12 322) were men, 81% (n=22 677) were of white ethnicity and 13% (n=3686) were South Asians (online supplemental table 1). The GENVASC population was, in general younger, had a higher proportion of Asians among the ethnic minorities, and had more participants from the extremes of the Townsend deprivation index (quintiles 1 and 5), compared with the NHSHC attendees across England or the general adult population aged between 40 and 74 years, although the differences were small (table 1).

### Cardiovascular risk groups

At recruitment, 18% (n=5090) of participants were classified to have high CVD risk and 4% (n=1162) to have very high CVD risk (online supplemental table 1). Patients in the high and very high CVD risk groups were on average older than patients in the low CVD risk groups (median age 65, 69 and 48 years, respectively), were more like to be male, of white ethnic origin and current smokers (online supplemental table 1).

### Diagnoses and risk factor detection

The majority of new clinical diagnoses of HTN, hypercholesterolaemia, diabetes mellitus and CKD occurred within the first 12 weeks following attendance to the NHSHC (figure 1). There was also a slight increase in the rate of new diagnoses of HTN, hypercholesterolaemia and diabetes between 12 weeks and 6 months post-NHSHC (figure 1). After that period the rate of new diagnoses plateaued and remained stable through the study (figure 1). In contrast to HTN, hypercholesterolaemia and diabetes, the rate of new diagnoses of atrial fibrillation was similar through the study duration.

There was a discrepancy between the proportion of diagnoses coded in medical records and observation of abnormal clinical and laboratory measurements (table 2). Within 12 weeks from recruitment to the NHSHC, abnormal BP measurements (SBP≥140 mm Hg and/or DBP≥90 mm Hg) were recorded for 27% (n=7516) individuals while coded diagnosis of HTN was recorded for only 2.3% (n=628). Similar pattern was observed for the

**Table 1** Comparison of GENVASC study participants included in the current analysis to the general population of England and large population of NHSHC attendees

| | ONS data* n=22 805 612 | NHSHC data* n=5 102 758 | GENVASC study n=27 888 | Missingness (%) |
|---|---|---|---|---|
| **Demographic** | | | | |
| Males, n (%) | 11 200 690 (49.1) | 2 311 604 (45.3) | 12 327 (44.3) | – |
| Age bands (years) | | | | – |
| 40–49 | 7 525 814 (33) | 1 951 264 (38.2) | 12 256 (43.9) | |
| 50–59 | 7 089 322 (31.1) | 1 742 003 (34.1) | 8495 (30.5) | |
| 60–74 | 8 190 476 (35.9) | 1 409 491 (27.7) | 7137 (25.6) | |
| Ethnicity, n (%) | | | | – |
| White | 20 383 677 (89.3) | 4 067 864 (79.7) | 22 677 (81.3) | |
| Asian | 1 341 580 (5.9) | 368 145 (7.2) | 3686 (13.2) | |
| Black | 585 756 (2.6) | 148 160 (2.9) | 535 (1.9) | |
| Other ethnicity | 494 599 (2.2) | 142 621 (2.8) | 990 (3.6) | |
| Missing data | | 375 968 (7.4) | | |
| Quintiles of deprivation, n (%) | | | | 2.4 |
| First quintile (least deprived) | 4 996 212 (21.9) | 1 129 670 (22.1) | 8332 (29.9) | |
| Second quintile | 4 901 834 (21.5) | 1 094 925 (21.5) | 5867 (21.0) | |
| Third quintile | 4 707 382 (20.6) | 1 027 096 (20.1) | 4429 (15.9) | |
| Fourth quintile | 4 286 645 (18.8) | 954 656 (18.7) | 3415 (12.3) | |
| Fifth quintile (most deprived) | 3 913 539 (17.2) | 893 194 (17.5) | 5167 (18.5) | |
| Missing data | – | 3217 (0.1) | 678 (2.4) | |

*Patel et al.[16]
†The presented Townsend scores were obtained using post codes provided by patients on consent forms (available for only a proportion of patients). All presented QRISK2 scores incorporate Townsend scores (available to GPs at the time of consultation).
GENVASC, The Genetics and Vascular Health Check; NHSHC, National Health Service Health Check; ONS, Office for National Statistics.

**Table 2** Comparison of coded diagnoses to rates of abnormal results with 12 weeks of the NHS Health Check

| Diagnoses made within 12 weeks from NHSHC | Counts (%) |
|---|---|
| Clinical diagnosis of HTN | 628 (2.3) |
| Abnormal BP reading (SBP≥140 mm Hg and/or DBP≥90 mm Hg) | 7516 (27) |
| Clinical diagnosis of hypercholesterolaemia | 70 (0.25) |
| High total cholesterol (TCh≥5 mmol/L) | 16 379 (58.7) |
| Very high total cholesterol (TCh≥7.5 mmol/L) | 547 (2) |
| Clinical diagnosis of diabetes mellitus | 248 (0.9) |
| Clinical diagnosis of obesity | 144 (0.5) |
| BMI meeting criteria for obesity (≥30 kg/m$^2$ and ≥27.5 kg/m$^2$ for Asian participants) | 7315 (26.2) |
| Clinical diagnosis of CKD | 19 (0.07) |
| Abnormal eGFR (<60 mL/min/1.73m$^2$) | 174 (0.9) |

BMI, body mass index; BP, blood pressure; CKD, chronic kidney disease; DBP, diastolic blood pressure; eGFR, estimated glomerular filtration rate; HTN, hypertension; NHSHC, National Health Service Health Check; SBP, systolic blood pressure; TCh, total cholesterol.

clinical diagnosis of hypercholesterolaemia, obesity and CKD. Particularly, 58.7% (n=16 379) and 2% (n=547) individuals were found to have TCh≥5 mmol/L and TCh≥7.5 mmol/L, respectively, while only 70 (0.25%) individuals received a coded diagnosis of hypercholesterolaemia. Coded diagnoses of obesity and CKD were recorded for 0.5% (n=144) and 0.07% (n=19) patients, respectively. However, within the same period values of BMI meeting criteria for obesity (≥30 kg/m$^2$ and ≥27.5 kg/m$^2$ for Asian participants) and eGFR (<60 mL/min/1.73m$^2$) were recorded for 26.2% (n=7315) and 0.9% (n=174) patients, respectively (table 2).

### Medical prescriptions and clinical interventions

Within the first 12 weeks following the NHSHC visit, 5.4% (1,510) of all attendees were started on antihypertensive medications, which increased to 8.4% (2337) at 18 months. Within the first 12 weeks from the NHSHC visit antihypertensive medications were prescribed for 3.6%, 10.0% and 21.2% of participants with low, high and very high CVD risk, respectively and 9.7%, 26.7% and 61.2% for grade 1, 2 and 3 HTN, respectively. At 18 months, these proportions increased to 6.4%, 14.8% and 26.2%, for the three CVD risk groups, and to 15.4%, 36.9% and 63.1% for the three HTN groups, respectively. At

18 months 4.5%, 13.5% and 28.8% of GENVASC participants, meeting criteria for grade 1, grade 2 and grade 3 HTN, respectively were prescribed more than one group of antihypertensive medications. The time from the NHSHC to onset of antihypertensive medication showed big discrepancies, however, there was a strong trend for shorter time to pharmacological treatment with higher category of HTN. In particular, median time from NHSHC to first prescription was reduced from 23 weeks for patients with grade 1 HTN, to 12 weeks for those with grade 2 HTN and 6 weeks for those with grade 3 HTN.

Similar trends were observed for the prescription of lipid lowering medications. Statins were prescribed to 5.4% (1514) of all attendees and the proportion gradually increased to 7.1% (1984) at 18 months. The proportion of patients prescribed statins increased with the severity of CVD risk, while the median time to statin prescription was decreased as the severity of CVD risk was increased. Specifically, within the first 12 weeks following the NHSHC, statins were prescribed to 1.8%, 14.2% and 34.2% of individuals in the low, high and very high CVD risk groups, respectively, reaching 2.8%, 18% and 39.2% at 18 months. The median time from the NHSHC to statin prescription was 29 weeks for the low CVD risk group, reduced to seven and 4 weeks for the high and very high CVD risk groups, respectively. Within the first 12 weeks following the NHSHC, 5.3% of participants with high TCh and 24.5% of participants with very high TCh were started on statins, reaching 7.4% and 33%, at 18 months, respectively.

### Frequency of follow-up

The frequency of follow-up visits was mainly associated with the severity of the abnormality detected during the NHSHC.

Over 18 months, patients, with grade 1, 2 and 3 HTN, as assessed during NHSHC visit, had a median of 4 (95% CI 2 to 9), 4 (95% CI 2 to 11) and 6 (95% CI 2 to 12) measurements of BP, respectively. Over 18 months of follow-up, at least a single repeated measurement of BP was performed in 41% of patients with normal, 58% with grade 1, 81% with grade 2 and 88.5% of patients with grade 3 HTN, respectively.

Over 18 months of follow-up repeated measurements of TCh were performed in 20%, 28% and 57% of patients with normal, high and very high TCh, respectively.

In contrast to the repeated measurements of BP and TCh, repeated measurements of body weight were done less frequently, with around one in four participants in total having a record of repeated weight measurements over the 18 months. At least a single repeated weight measurement was recorded for 17% of overweight and 25% of obese participants over 18 months of follow-up. There was a strong association between the overall cardiovascular risk and repeated measurements of BMI. At least on repeated measurement of BMI was performed for 24%, 30% and 37% of participants with low, high and very high CVD risk, respectively.

In addition to association with CVD risk there was an association between prescription of statins and antihypertensive medications and follow-up. Among patients with evidence of follow-up antihypertensive medications and statins were prescribed to 24% and 29.8% in comparison to 2.2% and 5.7% of patients with no evidence of follow-up (table 3).

### Change in the levels of CVD risk factors (first repeated measurement after 12 months from the NHSHC)

The magnitude of changes in the CVD risk factors during the follow-up period were relevant to the attendant's initial CVD risk severity group, allocated during the NHSHC visit, and were more evident in participants in the very high-risk group.

At follow-up participants in the low CVD risk category, had on average higher SBP and DBP by 3.6 (3.1 to 3.9) and 1.2 (1.0 to 1.6) mm Hg, respectively (average time to observation 74±15 weeks), and lower median eGFR by 5 (−5.6 to −4.3) mL/min/1.73m² (average time to observation 75±15 weeks). There was a minor reduction in the TCh levels by 0.14 (−0.16 to −0.10) mmol/L, (average time to repeated measurement 76±15 weeks) (table 4, low CVD group).

For participants in the high CVD risk group, there was no significant change in the SBP. DBP decreased by on average 0.9 (−1.4 to −0.40) mm Hg (average time to the follow-up measurement 73±15 weeks), while total and non-HDL cholesterol were decreased by 0.45 (−0.51 to −0.39) and 0.47 (−0.55 to −0.39) mmol/L, respectively (average time to repeated measurement 75±15 weeks). The median eGFR was lower by 3 (−4.4 to −1.6) mL/min/1.73m² (average time to repeated measurement 74±15 weeks). There was no significant change in the BMI at follow-up (table 4, high CVD group).

Participants in the very high CVD risk group, evidenced reduction, on average, in the SBP and DBP levels by 7.5 (−9.4 to −5.7) and 5 (−6.0 to −3.9) mm Hg, respectively (average time to repeated measurement 71±8 weeks) and in median eGFR by 3 (−5.0 to −0.8) mL/min/1.73m² (average time to repeated measurement 74±15 weeks). There were also clinically significant reductions in TCh and non-HDL cholesterol by 0.79 (−0.91 to −0.69) mmol/L, −0.68 (−0.83 to −0.53) mmol/L, respectively (average time to repeated measurement 74±15 weeks). The BMI reduced on average by −0.3 (−0.52 to −0.11) kg/m², respectively (average time to a repeated measurement 73±15 weeks) (table 4, very high CVD group).

Analogous to the gradient of CVD risk, the magnitude of the changes in the individual risk factors levels varied according the severity of the abnormality observed for that risk factor, during the NHSHC. Overall, individuals with BP above 140/90 mm Hg, as measured during the NHSHC, achieved an average reduction of 7 (−7.7 to −6.3) mm Hg of SBP and 4.6 (−5.1 to −4.2) mm Hg of DBP (average time to repeated observation 73±15 weeks). For grade 1, grade 2 and grade 3 HTN patients, SBP decreased on average by 2.9 (−3.7 to −2.3), 15.7 (−17.5

**Table 3** Demographic and clinical characteristics of participants with high and very high CVD risk who did not have record of follow-up

| | Follow-up | No follow-up | Delta (N-Y) (95% CI) | P value |
|---|---|---|---|---|
| High and very high CVD risk, n (%) (QRISK2≥10) | 4217 (67.5) | 2035 (32.5) | | |
| Demographic | | | | |
| Male sex, n (%) | 2738 (64.9) | 1378 (67.7) | 2.8 (0.3 to 5.2) | 0.0290 |
| White ethnic background, n (%) | 3615 (85.7) | 1850 (90.9) | 5.2 (3.5 to 6.8) | <0.0001 |
| Age, years (median, CI) | 64.2 (63.9 to 64.4) | 64.6 (64.4 to 64.9) | 0.47 (0.11 to 0.83) | 0.0104 |
| Quintiles of deprivation, n (%) | | | | |
| First quintile | 1240 (29.9) | 656 (32.8) | −2.9 (−5.4 to −0.4) | 0.0229 |
| Second quintile | 875 (21.1) | 483 (24.1) | −3.0 (−5.3 to 0.8) | 0.0077 |
| Third quintile | 707 (17.0) | 305 (15.2) | 1.8 (−0.1 to 3.7) | 0.0809 |
| Fourth quintile | 490 (11.8) | 236 (11.8) | – | 1 |
| Fifth quintile (most deprived) | 841 (20.2) | 323 (16.1) | 4.1 (2.0 to 6.2) | 0.0001 |
| Clinical | | | | |
| Current smokers, n (%) | 781 (18.5) | 359 (17.6) | −0.9 (−0.03 to 0.11) | 0.3876 |
| Ex-smokers, n (%) | 1176 (27.9) | 501 (24.6) | −3.3 (−5.6 to −0.9) | 0.0058 |
| BMI, kg/m$^2$ | 27.9 (5.2) | 26.9 (4.7) | −1.0 (−1.0 to −0.66) | <0.0001 |
| Systolic blood pressure, mm Hg | 139.3 (17.8) | 132.6 (13.5) | −6.7 (−7.0 to −5.9) | <0.0001 |
| Diastolic blood pressure, mm Hg | 82.3 (11.2) | 78.7 (8.9) | −3.6 (−4.0 to 3.1) | <0.0001 |
| Antihypertensive medications, n (%) | 1012 (24.0) | 44 (2.2) | −21.8 (−23.2 to −20) | <0.0001 |
| Total cholesterol, mmol/L | 5.6 (1.1) | 5.5 (1.0) | −0.1 (−0.1 to −0.02) | 0.0079 |
| Non-HDL cholesterol, mmol/L | 4.0 (1.0) | 3.9 (0.9) | −0.1 (−0.1 to 0.01) | 0.0622 |
| Statins, n (%) | 1257 (29.8) | 116 (5.7) | −24.1 (−25.8 to −22.4) | <0.0001 |

Results are mean (SD) unless otherwise stated. BP was defined as normal if SBP<140 mm Hg and/or DBP<90 mm Hg; TCh was defined as normal for values <5.0 mmol/L; non-HDL was defined as normal if values were <3.8 mmol/L; body weight was considered as healthy body weight if BMI<25 kg/m$^2$ (<23 kg/m$^2$ for South Asians).
BMI, body mass index; CVD, cardiovascular disease; HDL, high density lipoprotein; NHSHC, National Health Service Health Check.

to −14.1) and 33.4 (−37.7 to −29.4) mm Hg, respectively (figure 2, top panel).

A similar pattern was observed for the TCh levels, with an average reduction of 0.38 (−0.41 to −0.34) mmol/L and 1.71 (−1.94 to −1.48) mmol/L, respectively, for patients with high and very high TCh (average time to repeated measurement 75±15 weeks) (figure 2, middle panel).

There was a small reduction in the BMI with overweight patients losing approximately 0.08 (−0.15 to −0.007) kg/m$^2$ and obese patients losing 0.27 (−0.37 to −0.12) kg/m$^2$ (average time to repeated observation 75±15 weeks) (figure 2, bottom panel).

### Smoking cessation

During the NHSHC, 16% (3813; 19% men, 14% women) were recorded as current smokers. Of the 3813 current smokers, 56% (2133; 53% men, 47% women) had a record of smoking cessation advice documented during or within 30 days of the NHSHC. This proportion increased to 60% (2275) at 18 months, while 14% of participants had multiple records of advice for smoking cessation. At 18 months 451 (12%) participants quit smoking, according to the GP records.

### Individuals with no record of follow-up

The demographic and clinical characteristics of high and very high CVD risk patients with unrecorded follow-up are presented in table 3. Approximately one in three patients with high or very high CVD risk had no recorded evidence of a clinical follow-up in the 18-month period following the NHSHC. Lack of clinical follow-up was associated with male sex and white ethnic background and lower index of social deprivation (table 3). Participants without evidence of follow-up were more likely to be ex-smokers, had, at the time of the NHSHC, on average lower values of BMI, SBP and DBP than individuals who had record of follow-up, although the differences were small (table 3). Results from the logistic regression analysis showed that other than white ethnic background

**Table 4** Comparison of clinical characteristic of participants between NHSHC and follow-up

| Parameters | NHSHC | Follow-up (12–18 months) | Delta (FU-HC) (95% CI) | P value |
|---|---|---|---|---|
| Low CVD risk (QRISK2<10) | | | | |
| Obesity, n=3747 | | | | |
| Body mass index, kg/m² | 28.0 (5.9) | 28.0 (5.9) | 0.006 (−0.07 to 0.05) | 0.83 |
| Blood pressure, n=6785 | | | | |
| Systolic, mm Hg | 125.5 (15.6) | 129.1 (16.3) | 3.6 (3.1 to 3.9) | <0.001 |
| Diastolic, mm Hg | 78.7 (10.2) | 79.9 (10.4) | 1.2 (1.0 to 1.6) | <0.001 |
| Cholesterol, n=3233 | | | | |
| Total cholesterol, mmol/L | 5.4 (1.0) | 5.3 (0.98) | −0.14 (−0.16 to −0.10) | <0.001 |
| Non-HDL-cholesterol, n=1339 | | | | |
| Non-HDL-cholesterol, mmol/L | 3.9 (1.0) | 3.7 (0.98) | −0.12 (−0.16 to −0.08) | <0.001 |
| Median eGFR, n=2551 | | | | |
| eGFR, mL/min/1.73m² | 86 (85; 87) | 81 (75; 86) | −5 (−5.6 to −4.3) | <0.0001 |
| High CVD risk (QRISK2 10–20) | | | | |
| Obesity, n=1214 | | | | |
| Body mass index, kg/m² | 27.6 (5.4) | 27.6 (5.7) | −0.08 (−0.20 to 0.06) | 0.27 |
| Blood pressure, n=2046 | | | | |
| Systolic blood pressure, mm Hg | 136.7 (16.5) | 136.4 (16.6) | −0.3 (−1.1 to 0.53) | 0.5 |
| Diastolic blood pressure, mm Hg | 81.1 (10.8) | 80.2 (10.2) | −0.9 (−1.4 to −0.40) | <0.001 |
| Cholesterol, n=1339 | | | | |
| Total cholesterol, mmol/L | 5.6 (1.1) | 5.1 (1.1) | −0.45 (−0.51 to −0.39) | <0.001 |
| Non-HDL-cholesterol, n=583 | | | | |
| Non-HDL-cholesterol, mmol/L | 4.0 (1.0) | 3.6 (1.1) | −0.47 (−0.55 to −0.39) | <0.001 |
| Median eGFR, n=1129 | | | | |
| eGFR, mL/min/1.73 m² | 82 (78; 81) | 79 (78; 80) | −3 (−4.4 to −1.6) | <0.001 |
| Very high CVD risk (QRISK2>20) | | | | |
| Obesity, n=354 | | | | |
| Body mass index, kg/m² | 28.7 (6.1) | 28.4 (6.0) | −0.3 (−0.52 to −0.11) | 0.002 |
| Blood pressure, n=571 | | | | |
| Systolic blood pressure, mm Hg | 146.2 (20.8) | 138.7 (16.3) | −7.5 (−9.4 to −5.7) | <0.001 |
| Diastolic blood pressure, mm Hg | 84.8 (12.5) | 79.8 (10.4) | −5 (−6.0 to −3.9) | <0.001 |
| Cholesterol, n=419 | | | | |
| Total cholesterol, mmol/L | 5.6 (1.1) | 4.8 (1.2) | −0.79 (−0.91 to −0.69) | <0.001 |
| Non-HDL-cholesterol, n=174 | | | | |
| Non HDL-cholesterol, mmol/L | 4.1 (1.0) | 3.4 (1.1) | −0.68 (−0.83 to −0.53) | <0.001 |
| Median eGFR, n=303 | | | | |
| eGFR, mL/min/1.73m² | 81 (79; 83) | 78 (76; 80) | −3 (−5.0 to −0.8) | 0.006 |

Results are shown as mean (SD) unless otherwise indicated. Differences in mean values or proportions between the follow-up and the NHSHC, along with the relevant 95% CIs, are also shown. P values derived from the paired t-test for the continuous variables and the McNemar test for the nominal paired variables. BP was defined as normal if SBP<140 mm Hg and/or DBP<90 mm Hg; TCh was defined as normal for values <5.0 mmol/L; non-HDL was defined as normal if values were <3.8 mmol/L; kidney function was considered normal if eGFR≥60 mL/min/1.73m²; body weight was considered as healthy body weight if BMI<25 kg/m² (<23 kg/m² for South Asians).
CVD, cardiovascular disease; eGFR, estimated glomerular filtration rate; HDL, high density lipoprotein; NHSHC, National Health Service Health Check.

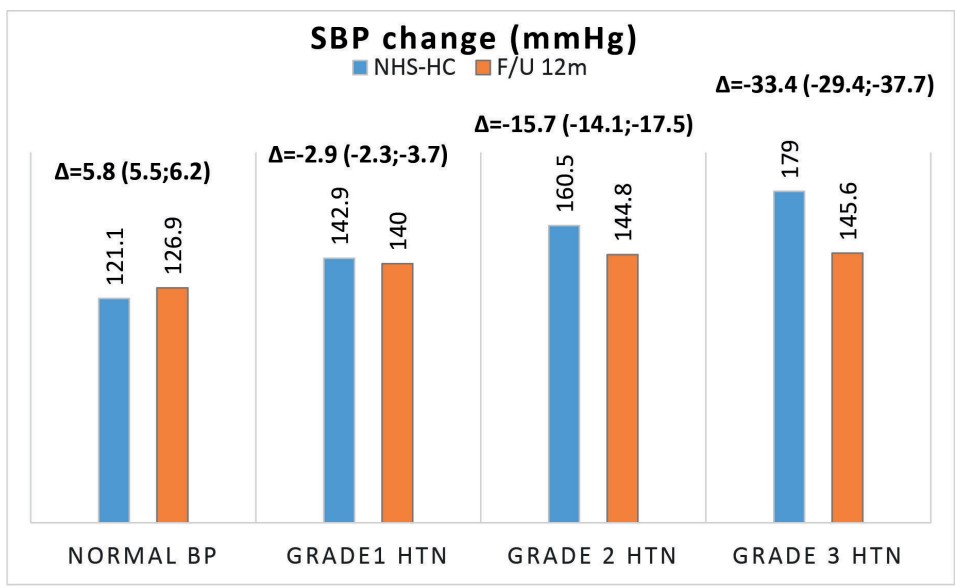

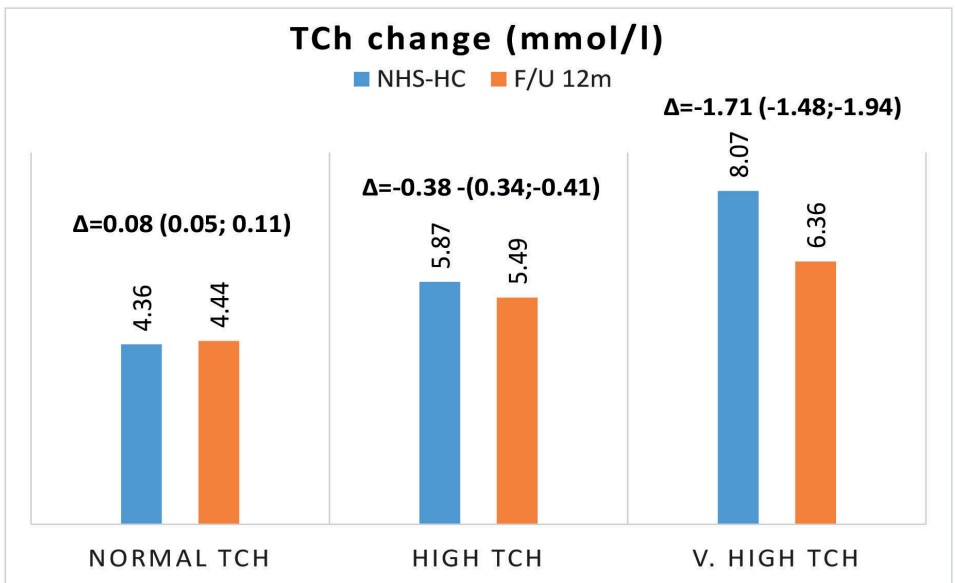

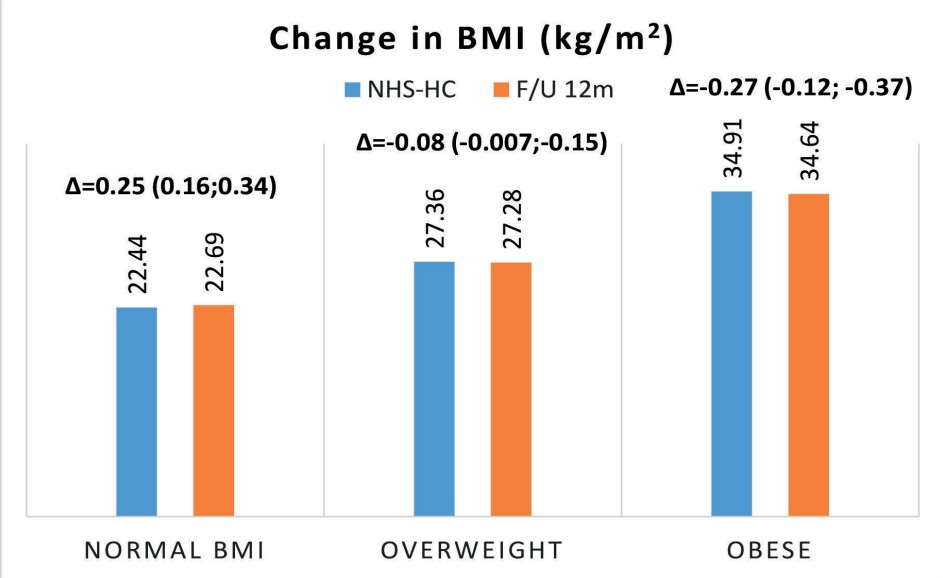

**Figure 2** Magnitude of changes in the individual risk factors during follow-up. BMI, body mass index; NHSHC, National Health Service Health Check; SBP, systolic blood pressure; TCh, total cholesterol.

(OR (95% CI) (1.79 (1.33 to 2.17), p<0.001), younger age (−1.02 (−1.06 to −1.001), p=0.004), greater SBP (1.02 (1.01 to 1.03), p<0.001), greater BMI (1.02 (1.01 to 1.04), p=0.001) and higher Townsend score (more deprived) (1.04 (1.01 to 1.06)), were independently associated with presence of follow-up record.

## DISCUSSION

We present results evaluating the clinical effectiveness of the NHSHC in identifying, treating and monitoring individuals at high CVD risk. Our study highlights novel findings which may influence implementation of strategies for improving effectiveness of the programme.

A key finding from our study is the characterisation of high and very high CVD risk patients (QRISK2≥10), who had attended the NHSHC, but have not had any record of clinical follow-up over 18 months and have not received any treatment. This is a considerably large subpopulation of patients and our study showed that this group has specific social, demographic and clinical characteristics. It is also the group of patients, where preventive interventions are likely to bring the largest clinical benefits and be most cost effective. Similar observations were made for patients with very abnormal values of isolated risk factors—such as BP and cholesterol. These subgroups of participants have a very high CVD risk, although it may not be accurately estimated by QRISK2.[13] This group of patients may need multidisciplinary approach using specialist services including genetic testing and cascade screening, which cannot be provided in primary care practices.

The second important finding relates to the level of control of risk factors. Despite the substantial improvements in both BP and TCh, their values at 18 months remained high (figure 2). This indicates that even when patients were started on treatment, there does not seem to be an imperative to ensure that the risk factors are well controlled. This may be due to several factors including high workload of primary care practices and lack of clear definition of primary prevention treatment targets. It is also likely that it reflects the way primary care is incentivised. Primary care practices receive a payment from NHS once a new case is identified or a new treatment is being initiated, rather than receiving a payment once treatment goals have been achieved.[21] Our study suggests that this mechanism may be insufficient. Our data also showed lack of adherence to the National Institute for Health and Care Excellence (NICE) recommendation, which advocates re-examination of lipid profile 3 months after prescribing statins.[14] Lack of repeated cholesterol measurements may be one of the factors significantly reducing uptake and continuation of statin treatment.[22]

Our findings are consistent with previous research indicating that attendance to NHSHC is associated with increased detection of CVD and its risk factors and prescription of therapy.[8 10 23] It also confirms the previous findings that the magnitude of the improvement in risk factors is relevant to the severity of the disease.[8 10] Based on the example of high BP management, our study provides strong evidence of the clinical effectiveness of NHSHC as a public programme aimed at reduction of CVD risk. However, the general perception of NHSC should be changed. NHSHC has to be recognised as a sequence of events starting with detection of increased CVD risk during the NHSHC, followed by general advice and prescription of therapy, and follow-up monitoring. Then a reassessment is required to decide if further changes to therapy are needed.

Our data indicate that the problem of patients with increased CVD risk (QRISK2≥10) not subjected to follow-up has not been adequately explored. A previous survey performed in general practices in London in the first year after implementation of the NHSHC identified problems with implementation.[24] The identified problems included non-prescribing of statins to high-risk individuals, reluctance to refer to external services and variable patterns of organising clinical follow-up with only around 50% of practices organising the recommended annual recall.[24] Forster *et al* also reported that fewer than 52% of patients had repeated monitoring of CVD risk factors over the 15-month period of observation.[8] Several studies reported lower than expected prescriptions of statins to patients diagnosed with elevated TCh and/or high CVD risk during NHSHC.[8–10]

Lower than projected uptake of the NHSHC has been a concern as the major factor influencing cost-effectiveness of the programme.[25] Recent data indicate a steady increase in the uptake now reaching a satisfactory level of 52%.[16 26] Our study indicates that overall effectiveness of the NHSHC can be improved by optimising delivery of interventions reducing the CVD risk.

### Limitations of the study

Our analysis has several limitations. Not all patients attending the NHSHC and invited to the GENVASC study took part and the exact proportion (uptake) is not known. It is possible that the patients who participated in the study were more proactive in attending follow-up appointments and undertaking medical and lifestyle interventions than the general population of attenders to the NHSHC. Our analysis was performed in a population with higher than the average for UK proportion of Asian participants. The performed analysis focused on CVD risk factors recorded in the primary care records but no data were available regarding lifestyle changes (eg, on physical exercise, diet or alcohol consumption). Furthermore, we do not have data on prescription of antiobesity drugs, or referral to weight loss, alcohol or diabetes prevention services.

**Author affiliations**
[1]Department of Cardiovascular Sciences and NIHR Cardiovascular Research Centre, University of Leicester, Leicester, UK
[2]Department of Food Science and Nutrition, University of the Aegean, Lemnos, Greece
[3]Albany House Medical Centre, Wellingborough, UK

[4]Diabetes Research Centre, University of Leicester, Leicester, UK

[5]Diabetes and Cardiovascular Medicine General Practice Alliance Federation Research and Training Academy, Northampton, UK

[6]South Leicestershire Medical Group, Kibworth Beauchamp, UK

[7]Lakeside Healthcare Research, Corby, UK

[8]Willowbrook Medical Centre, Leicester, UK

[9]Department of Health Sciences, University of Leicester, Leicester, UK

[10]Institute of Cardiovascular Science, University College London, London, UK

**Acknowledgements** The research team would like to express gratitude to all participant of the GENVASC study. We would also like to thank all healthcare and other professionals involved in performing NHS Health Check as well as research related tasks.

**Contributors** RD, RP, NJS: designed the project, performed analysis and are gurantors and are responsible for the overall contents of the manuscript. VB: designed the project, performed analysis, supervised the statistical analysis and is responsible for the overall contents of the manuscript. DL, CG, RB, SS: took part in collection and analysis of data, was involved in managing of the research databases and download of data, reviewed the content and contributed to the final manuscript. EB: took part in collection and analysis of data, was involved in overall management of the study, reviewed the content and contributed to the final manuscript. SK, AK, AZ, AC, AH, KK, NJ, ML, AF: took part in collection and analysis of data, reviewed the content and contributed to the final manuscript. MN: took part in collection and analysis of data, was involved in and overlooked clinical governance of the project, reviewed the content and contributed to the final manuscript. AM: took part in collection and analysis of data, was involved in overall management of the study, reviewed the content and contributed to the final manuscript.

**Funding** The GENVASC study is funded by the NIHR Leicester Biomedical Research Centre (grant no. BRC-1215-20010). RD is funded by NIHR (grant number NA). KK is supported by the National Institute for Health Research (NIHR) Applied Research Collaboration East Midlands (ARC EM) and the NIHR Leicester Biomedical Research Centre (BRC) (grant number NA).

**Map disclaimer** The inclusion of any map (including the depiction of any boundaries therein), or of any geographic or locational reference, does not imply the expression of any opinion whatsoever on the part of BMJ concerning the legal status of any country, territory, jurisdiction or area or of its authorities. Any such expression remains solely that of the relevant source and is not endorsed by BMJ. Maps are provided without any warranty of any kind, either express or implied.

**Competing interests** KK was a national advisor for the NHS Health Checks programme.

**Patient and public involvement** Patients and/or the public were involved in the design, or conduct, or reporting, or dissemination plans of this research. Refer to the Methods section for further details.

**Patient consent for publication** Not applicable.

**Ethics approval** This study involves human participants and was approved by the East Midlands - Derby Research Ethics Committee (approval number 12/EM/0208). All study subjects gave informed consent for participation.

**Provenance and peer review** Not commissioned; externally peer reviewed.

**Data availability statement** Data are available upon reasonable request. Summary data available on request from authors, subject to privacy/ethical restrictions.

**ORCID iDs**
Radoslaw Debiec http://orcid.org/0000-0003-2292-467X
Anuj Chahal http://orcid.org/0000-0001-5664-4487
Riyaz Patel http://orcid.org/0000-0003-4603-2393

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
