## [Reviewer comments · BMJ Open]

ARTICLE DETAILS

TITLE (PROVISIONAL)	Evaluating the clinical effectiveness of the NHS Health Check programme: A prospective analysis in the Genetics and Vascular Health Check (GENVASC) Study
AUTHORS	Debiec, Radoslaw; Lawday, Daniel; Bountziouka, Vasiliki; Beeston, Emma; Greengrass, Chris; Bramley, Richard; Sehmi, Sue; Kharodia, Shireen; Newton, Michelle; Marshall, Andrea; Krzeminski, Andre; Zafar, Azhar; Chahal, Anuj; Heer, Amardeep; Khunti, Kamlesh; Joshi, Nitin; Lakhani, Mayur; Farooqi, Azhar; Patel, Riyaz; Samani, Nilesh

VERSION 1 – REVIEW

REVIEWER	Harvie, Michelle University Hospital of South Manchester NHS Foundation Trust, Genesis prevention centre
REVIEW RETURNED	18-Dec-2022

GENERAL COMMENTS	Study overview This is an interesting analysis of outcomes of NHS health checks in a large population undergoing a health check including follow up data and examination of key socio-economic factors which are associated with follow up. A useful addition to UK NHS health check data. Overall comments  • The title and introduction would be more accurate to describe this as an analysis within the GENVASC study rather than a study itself. The GENVASC study aims to test the addition of PRS scores to Q RISK for CVD risk prediction and not primarily the outcomes described herein • Need to include trial registration of the GENVASC study Study participants  • Useful to have uptake to the GENVASC study to provide a sense any likely bias in the study population • Would be clearer to say included patients from the study recruited 17/06/2010 and 04/09/2018 who had 18-month FUP data for the current analysis. Outcomes of interest Needs to include smoking as these are reported in the results. Is there any data on physical activity, alcohol. Assuming there is not data, this is a limitation of the trial which should be mentioned in the discussion Analysis How did the researchers deal with patients who had died before 12/ 18 months of follow up? Results  • Table 1 Please check the figures as the totals don't add up
---

- Table 2 Please add QRISK cut offs for low high / v high risk and the cut offs for the levels of hypertension, cholesterol etc
- Figure 1 Weekly incidence of CVDs in relation to NHS Health Check – should be renamed as CVD risk factors
- What are the differences in obesity and obesity based on BMI. Also, what are the differences between chronic kidney disease and chronic kidney disease based on EGFR < 60?
- Supplementary Table 1 why is there 43.5 % missingness for quintiles of deprivation scores. This should be discussed as a limitation. These should be available from a post code. This is a concern as these are also part of the QRISK score reported.

Diagnosis and risk factor detection

There is quite a lot of information here including different ways that HTN, diabetes etc was reported after the health check. It would be clearer to include this in a simple table, if possible

This section includes results for prescription of statins etc which is repeated in the Medical Prescriptions and clinical intervention section. Suggest this is removed from here.

Medical Prescriptions and clinical interventions

- Suggest saying 18 months throughout this section rather than end of the study to help the reader
- Were there any prescriptions for weight loss medicines / referral to tier 2/3/4 weight loss services?
- Is there any data on commencing appropriate medications for atrial fibrillation which is a target of the NHS health checks ?
- Any data on referrals to alcohol services ?
- Any data on referral to the diabetes prevention programme for hyperglycaemia?

Frequency of follow-up

- Need to have the same order of outcomes in each of these results sections i.e., HTN, hypercholesterolaemia etc throughout all sections to help the reader. Some have cholesterol first.
- Were there any differences in follow up for patients on medications vs those who were not?
- Were weights repeated only in patients with high cholesterol/ BP on medications?

Change in the levels of CVD risk factors

- It would be useful to remind the reader the time-period the changes were measured between i.e., mean (SD) for time for the different measures.
- Should report mean and 95% confidence intervals in the text rather than just the average change in the text
- Not sure how useful this sentence is – suggest removing as better explained by the three hypertension groups
“Overall, individuals with BP above 140/90 mmHg, as measured during the NHSHC, achieved an average reduction of 7mmHg of SBP and 5mmHg of DBP after 12 months from assessment.”

- The results for weight are confusing. Graphs and sentence one look like there are no difference, the greater reductions in patients with obesity vs overweight represent v small changes which are unlikely to be clinically significant.

“This trend was not observed for body weight. In particular, there was a small difference in the magnitude of reduction of BMI over the follow up period with overweight patients losing approximately 0.1kg/m2 and obese patients losing

	on average 0.3 kg/m². (Figure 2)” Individuals with no record of follow-up  • The definition of follow up is based on repeat measurements. However, this group includes patients started were started on medications. Need to highlight that lack of follow up was more common in those prescribed statins and anti- hypertensive medications as this is important for interpretation of the data. • This results section could be more succinct. • Table 3 headings could be clearer. Suggest follow up / no follow up as headings. • Add % to the numbers on the follow up and no follow up groups • The term lost in the follow up column is confusing. These patients are still in the study and not lost to follow up. Their outcome is that they were either followed up or not • Were use of statin and anti-hypertensives included in the regression? Discussion The discussion should be more concise, and include strength and limitation, relevance to current practice and future work. The discussion includes some results on patients with isolated risk factors i.e., bp / cholesterol. Whilst this makes an important point, suggest the data is in the results and the related discussion is in the discussion. The paragraph on incentives is an important point but could be made more succinct Some paragraphs are not clear  1. This may be due to several factors including high workload of GP practices and the divide between primary and secondary prevention, where for the latter, the targets are much better defined in guidelines Would be clearer to say need better prevention targets as there are for secondary prevention.  2. “Lack of repeated cholesterol measurements may be one of the factors significantly reducing uptake and continuation of statin treatment. This statement needs justification or a reference to justify as sounds a bit speculative. Do we know that feedback on cholesterol levels is associated with adherence to statins?  3. This paragraph is unclear. Is this specifically related to moderate and high risk rather than very high-risk patients. Our data indicate that the problem of patients with moderate and high CVD risk not subjected to follow-up has not been adequately explored. A previous survey study performed in general practices in London in the first year after implementation of the NNSHC identified problems with implementation.²⁴ The identified problems included non-prescribing of statins to high-risk individuals, reluctance to refer to external services and variable patterns of organising clinical follow-up with only around 50% of practices organising the recommended annual recall.²⁴ Forester et al. (2015) also reported that fewer than 52% of patients had repeated monitoring of CVD risk factors over the 15-month period of observation.⁸ Several studies reported lower than expected prescriptions of statins to patients diagnosed with elevated TCh and/or high CVD risk during NNSHC.⁸⁻¹⁰ The final paragraph is a bit off topic, need to say whilst uptake has improved there needs to be more focus on downstream activity Other limitations Lack of lifestyle outcomes and lifestyle referrals which is a potential issue of the NNSHC programme which is mainly focussing on medication prescription
--	--

VERSION 1 – AUTHOR RESPONSE

Reviewer: 1

Dr. Michelle Harvie, University Hospital of South Manchester NHS Foundation Trust

Comments to the Author:

Study overview

This is an interesting analysis of outcomes of NHS health checks in a large population undergoing a health check including follow up data and examination of key socio-economic factors which are associated with follow up. A useful addition to UK NHS health check data.

Overall comments

- The title and introduction would be more accurate to describe this as an analysis within the GENVASC study rather than a study itself. The GENVASC study aims to test the addition of PRS scores to Q RISK for CVD risk prediction and not primarily the outcomes described herein

The title of the manuscripts was amended:

Was: Clinical effectiveness of the NHS Health Check programme: Insights from the Genetics and Vascular Health Check (GENVASC) Study

Is: Evaluating the clinical effectiveness of the NHS Health Check programme: A prospective analysis in the Genetics and Vascular Health Check (GENVASC) Study

- Need to include trial registration of the GENVASC study

ClinicalTrials.gov Identifier: NCT04417387 was added in Introduction- See "Introduction" lines 101-102.

Study participants

- Useful to have uptake to the GENVASC study to provide a sense any likely bias in the study population

The accurate data on the uptake to the study is not available. Due to the scale of the project and dispersed recruitment of participants across almost 150 practices, there was no systematic screening log to document individuals who were invited to participate but declined. The information about this limitation has now been added to the study limitations paragraph in the Discussion (lines 468-477).

- Would be clearer to say included patients from the study recruited 17/06/2010 and 04/09/2018 who had 18-month FUP data for the current analysis.

The first sentence of the “Study Participants” paragraph was amended. See “Study Participants” lines 141-142

Was:

Our study sample comprised of 27,888 participants of the GENVASC study recruited between 17/06/2010 and 04/09/2018. This time limits were set to allow for 18 months of follow-up after the initial NHSHC.¹⁵

Is:

For the purpose of current analysis we included 27,888 participants of the GENVASC study recruited between 17/06/2010 and 04/09/2018 who had 18 months of follow-up data.¹⁵

Outcomes of interest

Needs to include smoking as these are reported in the results. Is there any data on physical activity, alcohol. Assuming there is not data, this is a limitation of the trial which should be mentioned in the discussion

Thank you. The paragraph “Outcomes of interest” was amended. See “outcomes of interest lines 174-184.

Was:

- risk factor detection - absolute number and proportion of subjects diagnosed with an abnormal values of BP, TCh, eGFR, BMI or having a clinical diagnosis of HTN, diabetes, hypercholesterolaemia, CKD or atrial fibrillation
- medication prescription - absolute number and proportion of subjects started on a given therapy

Is:

- risk factor detection - absolute number and proportion of participants with an abnormal values of BP, TCh, eGFR, BMI, current tobacco smoking or having a clinical diagnosis of HTN, diabetes, hypercholesterolaemia, CKD, or atrial fibrillation. Both the rates of clinically coded diagnoses and the rates of observed abnormal values were presented to accurately assess prevalence of risk factors.
- medication prescription and non-medical interventions (e.g. smoking cessation) - absolute number and proportion of subjects started on a given therapy

A sentence was added to the paragraph “Strengths and limitations” of the study. See lines 84-86 and paragraph “Limitations of the study lines 468-477.

Analysis

How did the researchers deal with patients who had died before 12/ 18 months of follow up?

The current analysis was performed on a preselected group of patients with available 18 months follow up data. Patients who died within 18 months from the NHSHC were not included in the analysis.

See: Paragraph “study Participants”, lines 141-142.

For the purpose of current analysis we included 27,888 participants of the GENVASC study recruited between 17/06/2010 and 04/09/2018 who had 18 months of follow-up data.¹⁵

Results

Table 1 Please check the figures as the totals don't add up.

Authors thank the reviewer for the comment.

The numbers in the Table1 have been cross-checked with the source and amended.

Table 2 Please add QRISK cut offs for low high / v high risk and the cut offs for the levels of hypertension, cholesterol etc

Thank you. QRISK2 cut-offs for relevant risk categories are now added in Tables. The legend of Tables have been supplemented with the cut-offs for normal values of presented variables.

- Figure 1 Weekly incidence of CVDs in relation to NHS Health Check – should be renamed as CVD risk factors

Title of Figure 1 was amended. Thank you

Was:

Weekly incidence of CVDs in relation to NHS Health Check

Is:

Weekly incidence of CVD risk factors in relation to NHS Health Check

- What are the differences in obesity and obesity based on BMI. Also, what are the differences between chronic kidney disease and chronic kidney disease based on EGFR < 60?

The data show that clinic records under-report prevalence of risk factors e.g. only a proportion of patients with abnormal eGFR values receive a coded diagnosis of chronic kidney disease.

An explanatory sentence was added in the paragraph “Outcome of interest” lines 176-178.

Both the rates of clinically coded diagnoses and the rates of observed abnormal values are presented to accurately assess prevalence of risk factors.

Further explanation is also provided in the paragraph “Diagnoses and risk factor detection” Lines 264-275

- Supplementary Table 1 why is there 43.5 % missingness for quintiles of deprivation scores. This

should be discussed as a limitation. These should be available from a post code. This is a concern as these are also part of the QRISK score reported.

Thank you very much for your comment and highlighting a very significant problem with the data. The QRISK2 scores used in the analysis were calculated by General Practitioners during the NHSHC visit and incorporate Townsend score. However, Townsend deprivations scores were not recorded in Primary Care records.

The Townsend scores presented in the tables were obtained using post codes provided by patients on consent forms. Following the concern raised by the reviewer the team has investigated the missing data.

It was noted the post codes recorded in the database after the date of the NHSHC were being rejected.

Following the recruitment of the patients, the consent forms were being sent to the research unit and subsequently manually inserted into the database. Therefore, there were differences between the dates of NHS Health Check and the dates post codes were recorded (the differences were results of delays between obtaining the consent and manual input of the post codes in the database). Thus, the criterion to reject post codes with dates recorded after the NHSHC was overly and unnecessarily restrictive. It has been altered to include all post codes recorded on consent forms.

Following the alternation of the script post codes (derived from consent form obtained during recruitment) were available for 99.7% of participants. Based on the available post codes Townsend scores were obtained for 97.6%.

The research team re-calculated statistics using the Townsend scores and updated the tables.

Diagnosis and risk factor detection

- There is quite a lot of information here including different ways that HTN, diabetes etc was reported after the health check. It would be clearer to include this in a simple table, if possible

Thank you. The paragraph has been modified and a Table with results added. See Table 2.

Diagnoses and risk factor detection

The majority of new clinical diagnoses of HTN, hypercholesterolaemia, diabetes mellitus and CKD occurred within the first 12 weeks following attendance to the NHSHC. (Figure 1) There was also a slight increase in the rate of new diagnoses of HTN, hypercholesterolaemia and diabetes between 12 weeks and six months post NHSHC. (Figure 1) After that period the rate of new diagnoses plateaued and remained stable through the study. (Figure 1) In contrast to HTN, hypercholesterolaemia and diabetes, the rate of new diagnoses of atrial fibrillation was similar through the study duration.

There was a discrepancy between the proportion of diagnoses coded in medical records and observation of abnormal clinical and laboratory measurements. (Table 2) Within 12 weeks from recruitment to the NHSHC, abnormal BP measurements (SBP \geq 140mmHg and/or DBP \geq 90mmHg) were recorded for 27% (n=7,516) individuals whilst coded diagnosis of HTN was recorded for only 2.3% (n=628). Similar pattern was observed for the clinical diagnosis of hypercholesterolaemia, obesity, and CKD. Particularly, 58.7% (n=16,379) and 2% (n=547) individuals were found to have TCh \geq 5mmol/l and TCh \geq 7.5mmol/l, respectively, whilst only 70 (0.25%) individuals received a coded diagnosis of hypercholesterolaemia. Coded diagnoses of obesity and CKD were recorded for 0.5% (n=144) and 0.07% (n=19) patients, respectively. However, within the same period values of BMI meeting criteria for obesity (\geq 30 kg/m² and \geq 27.5 kg/m² for Asian participants) and eGFR (<60

ml/min/1.73m²) were recorded for 26.2% (n=7,315) and 0.9% (n=174) patients, respectively. (Table 2)

Table 2 Comparison of coded diagnoses to rates of abnormal results with 12 weeks of the NHS Health Check

Diagnoses made within 12 weeks from NHSHC	Counts (%)
Clinical diagnosis of HTN	628 (2.3)
Abnormal BP reading (SBP≥140mmHg and/or DBP≥90mmHg)	7,516 (27)
Clinical diagnosis of hypercholesterolaemia	70 (0.25)
High total cholesterol (TCh ≥5mmol/l)	16,379 (58.7)
Very high total cholesterol (TCh ≥7.5mmol/l)	547 (2)
Clinical diagnosis of diabetes mellitus	248 (0.9)
Clinical diagnosis of obesity	144 (0.5)
BMI meeting criteria for obesity (≥30 kg/m ² and ≥27.5 kg/m ² for Asian participants)	7,315 (26.2)
Clinical diagnosis of CKD	19 (0.07)
Abnormal eGFR (<60ml/min/1.73m ²)	174 (0.9)

NHSHS – NHS Health Check, HTN – hypertension, BP – Blood pressure, SBP – Systolic blood pressure, DBP – Diastolic blood pressure TCh – total cholesterol, BMI – body mass index, CKD – chronic kidney disease, eGFR – estimated glomerular filtration rate.

- This section includes results for prescription of statins etc which is repeated in the Medical Prescriptions and clinical intervention section. Suggest this is removed from here.

The information about prescription of antihypertensive medications and statins has been removed from the paragraph. Thank you for this suggestion

Medical Prescriptions and clinical interventions

- Suggest saying 18 months throughout this section rather than end of the study to help the reader

In the paragraph and the rest of the manuscript (where feasible) the terms “end of the study” and “end of the study duration” were replaced by “18 months

- Were there any prescriptions for weight loss medicines / referral to tier 2/3/4 weight loss services?

These data are currently not available. Relevant comment was added to the section regarding limitations if the study.

- Is there any data on commencing appropriate medications for atrial fibrillation, which is a target of the NHS health checks?

We thank the Reviewer for their comment.

Prescription of anticoagulation follows complex clinical algorithm which includes calculation of risk of stroke and calculation of bleeding risk. The clinical decision of prescribing this medication is based on these factors. We had no access to the GP calculated stroke and bleeding risk scores, therefore, presenting prescription data without relevant risk scores could be misleading. Moreover, unlike the

other risk factors, the rate of new diagnosis for atrial fibrillation showed no association with the NHS Health Check in our data.

- Any data on referrals to alcohol services?

These data are currently not available. Relevant comment was added to the section regarding limitations if the study.

- Any data on referral to the diabetes prevention programme for hyperglycaemia?

These data are currently not available. Relevant comment was added to the section regarding limitations if the study.

Frequency of follow-up

- Need to have the same order of outcomes in each of these results sections i.e., HTN, hypercholesterolaemia etc throughout all sections to help the reader. Some have cholesterol first.

The order of outcomes in the paragraph has now changed to follow the same order as in other paragraphs.

- Were there any differences in follow up for patients on medications vs those who were not?

The information on association of medicines prescription with follow up is presented in Table 4.

There was a significant association between prescription of statins and anti-hypertensive medications and follow up. Among patients with evidence of follow up antihypertensive medications and statins were prescribed to 24% and 29.8% in comparison to 2.2% and 5.7 % of patients with no evidence of follow up.

- Were weights repeated only in patients with high cholesterol/ BP on medications?

We thank the Reviewer for the comment. The relationship between repeated weight measurement and cardiovascular risk had not been previously explored.

A sentence was added to the paragraph "Frequency of follow-up" Lines 325-327

There was a strong association between the overall cardiovascular risk and repeated measurements of BMI. At least one repeated measurement of BMI was performed for 24%, 30% and 37% of participants with low, high and very high CVD risk, respectively.

Change in the levels of CVD risk factors

- It would be useful to remind the reader the time-period the changes were measured between i.e., mean (SD) for time for the different measures.

The Title of the paragraph has been amended:

Was:

Change in the levels of CVD risk factors

Is:

Change in the levels of CVD risk factors (first repeated measurement after 12 months from the NHSHC)

The paragraph was amended to provide average time to repeated measurement (mean+/-SD)

- Should report mean and 95% confidence intervals in the text rather than just the average change in the text

The 95% CI have now been added within the text.

- Not sure how useful this sentence is – suggest removing as better explained by the three hypertension groups

“Overall, individuals with BP above 140/90 mmHg, as measured during the NHSHC, achieved an average reduction of 7mmHg of SBP and 5mmHg of DBP after 12 months from assessment.”

Authors agree the change in SBP stratified as per three hypertension groups are more informative for the reader than the average change for all hypertensive patients. However, presentation of the average reduction in SBP for all hypertensive individuals aims to enable comparison to previous publications reporting this outcome (e.g. Forester et al. PMID: 25326192) and can be used for health economics analysis.

- The results for weight are confusing. Graphs and sentence one look like there are no difference, the greater reductions in patients with obesity vs overweight represent v small changes which are unlikely to be clinically significant.

The paragraph describing change in BMI during follow up has been amended as follows: (Lines 381-383)

Was:

“This trend was not observed for body weight. In particular, there was a small difference in the magnitude of reduction of BMI over the follow up period with overweight patients losing approximately 0.1kg/m² and obese patients losing on average 0.3 kg/m². (Figure 2)”

Is:

There was a small reduction in the BMI with overweight patients losing approximately 0.08 (-0.15; -0.007) kg/m² and obese patients losing 0.27 (-0.37; -0.12) kg/m² (average time to repeated observation 75+/-15 weeks). (Figure 2, bottom panel)

- The definition of follow up is based on repeat measurements. However, this group includes patients started were started on medications. Need to highlight that lack of follow up was more common in those prescribed statins and anti- hypertensive medications as this is important for interpretation of the data.

Thank you. The requested information was supplemented in the paragraph “Frequency of follow-up” Lines 328-331

In addition to association with CVD risk there was an association between prescription of statins and anti-hypertensive medications and follow up. Among patients with evidence of follow up antihypertensive medications and statins were prescribed to 24% and 29.8% in comparison to 2.2% and 5.7 % of patients with no evidence of follow up. (Table 4)

- This results section could be more succinct.

The authors made effort to condense the section while supplementing missing information

- Table 3 headings could be clearer. Suggest follow up / no follow up as headings.

The Heading (now table 4) were changed to “Follow up”, “No follow up”

- Add % to the numbers on the follow up and no follow up groups

Thank you. Counts of patients with and without follow up have now been supplemented with proportions (%).

- The term lost in the follow up column is confusing. These patients are still in the study and not lost to follow up. Their outcome is that they were either followed up or not

The term “lost to follow up” was removed.

- Were use of statin and anti-hypertensives included in the regression?

Authors thank the reviewer for the important comment.

In a univariate analysis (Table 4) higher proportion of patients attending the follow up received prescriptions for statins and anti-hypertensive medications. The authors have not included prescription of statins and anti-hypertensive medication in the logistic regression to avoid misinterpretation of results. Prescription of medications could be a factor influencing decision about follow up i.e. GPs prescribing medications to a patient are more/less worried about the CVD risk in that patient which influences the decision about the follow up. However, an alternative explanation is that GPs would not prescribe medications because patients do not attend follow up.

Discussion

The discussion should be more concise, and include strength and limitation, relevance to current practice and future work.

The authors thank the reviewer for the insightful comments. The authors made amendments to discussion to make it more concise and highlight limitations which were previously not mentioned. This in authors' opinion led to stronger overall message of the manuscript.

The discussion includes some results on patients with isolated risk factors i.e., bp / cholesterol. Whilst this makes an important point, suggest the data is in the results and the related discussion is in the discussion.

The authors removed the references to specific results.

The paragraph on incentives is an important point but could be made more succinct

The paragraph was shortened and made more transparent.

Some paragraphs are not clear

1. This may be due to several factors including high workload of GP practices and the divide between primary and secondary prevention, where for the latter, the targets are much better defined in guidelines

Would be clearer to say need better prevention targets as there are for secondary prevention.

The relevant statement was amended:

Was: This may be due to several factors including high workload of GP practices and the divide between primary and secondary prevention, where for the latter, the targets are much better defined in guidelines

Is: This may be due to several factors including high workload of GP practices and lack of clear definition of primary prevention treatment targets. Lines 432-434.

2. "Lack of repeated cholesterol measurements may be one of the factors significantly reducing uptake and continuation of statin treatment. This statement needs justification or a reference to justify as sounds a bit speculative. Do we know that feedback on cholesterol levels is associated with adherence to statins?"

A relevant reference was added.

Benner JS, Tierce JC, Ballantyne CM, Prasad C, Bullano MF, Willey VJ, Erbey J, Sugano DS. Follow-up lipid tests and physician visits are associated with improved adherence to statin therapy. *Pharmacoeconomics* 2004;22 Suppl 3:13-23.

3. This paragraph is unclear. Is this specifically related to moderate and high risk rather than very high-risk patients.

The authors thank the reviewer for the comment.

NICE guidelines* on primary prevention of CVD recommend clinical intervention to reduce CVD risk for all individuals with QRISK2 ≥ 10 . This is relevant for both high and very high CVD risk groups in our study.

To make the paragraph more transparent the authors amended the first sentence in the paragraph. Lines 451-452.

Was:

Our data indicate that the problem of patients with moderate and high CVD risk not subjected to follow-up has not been adequately explored.

Is:

Our data indicate that the problem of patients with increased CVD risk (QRISK2 ≥ 10) not subjected to follow-up has not been adequately explored.

*"Cardiovascular disease: risk assessment and reduction, including lipid modification"
, <https://www.nice.org.uk/guidance/cg181/chapter/Recommendations#identifying-and-assessing-cardiovascular-disease-cvd-risk>

The final paragraph is a bit off topic, need to say whilst uptake has improved there needs to be more focus on downstream activity

The last paragraph of the discussion was amended. Lines 461-465.

Was:

Lower than projected uptake of the NHSHC has been a concern as the major factor influencing cost-effectiveness of the programme.²⁵ The reason for non-attendance to NHSHC has been extensively investigated using qualitative and quantitative approaches. Recent data indicate steady increase in the uptake now reaching 52%.¹⁶ This threshold of 50% has been commented in the context of sensitivity analysis as satisfactory.²⁶ Our study indicates a new goal for the research community and Public Health England to work on in the pursuit of improving the overall effectiveness of thNHSHC.

Is:

Lower than projected uptake of the NHSHC has been a concern as the major factor influencing cost-effectiveness of the programme.²⁶ Recent data indicate steady increase in the uptake now reaching a satisfactory level of 52%.^{16, 27} Our study indicates that overall effectiveness of the NHSHC can be improved by optimising delivery of interventions reducing the CVD risk.

Lack of lifestyle outcomes and lifestyle referrals which is a potential issue of the NHSHC programme which is mainly focussing on medication prescription

A paragraph about limitation of the study has now been added. Lines 467-477

Is there data on ongoing prescription information?

GENVASC is a study with an ongoing collection of data on cardiovascular outcomes, risk factors, follow up including ongoing medicine prescription. The 18 months follow up period was chosen to enable analysis of large proportion of study participants over a period which was considered sufficiently long to assess the effectiveness of the clinical interventions introduced at the time of the NNSHC.